# Outcomes of the Health Insurance Card Scheme on Migrants’ Use of Health Services in Ranong Province, Thailand

**DOI:** 10.3390/ijerph17124431

**Published:** 2020-06-19

**Authors:** Mathudara Phaiyarom, Nareerut Pudpong, Rapeepong Suphanchaimat, Watinee Kunpeuk, Sataporn Julchoo, Pigunkaew Sinam

**Affiliations:** 1International Health Policy Program, Ministry of Public Health, Nonthaburi 11000, Thailand; nareerut@ihpp.thaigov.net (N.P.); rapeepong@ihpp.thaigov.net (R.S.); watinee@ihpp.thaigov.net (W.K.); sataporn@ihpp.thaigov.net (S.J.); pigunkaew@ihpp.thaigov.net (P.S.); 2Division of Epidemiology, Department of Disease Control, Ministry of Public Health, Nonthaburi 11000, Thailand

**Keywords:** migrant, service utilisation, health insurance, Health Insurance Card Scheme, Thailand

## Abstract

In 2002, Thailand achieved Universal Health Coverage for all citizens; however, it remains the case that undocumented migrants are not fully covered. The Health Insurance Card Scheme (HICS) of the Ministry of Public Health is the key policy aiming to cover undocumented migrants. This study examined the impact of this policy on the utilisation rate of public health facilities among HICS beneficiaries including undocumented migrants. Facility-based individual records between 2011 and 2015 were purposively retrieved from one provincial hospital, one district hospital, and two health centres in one of the most densely migrant-populated provinces in Thailand. Poisson regression was conducted on inpatient (IP) utilisation, while negative binomial regression was conducted on outpatient (OP) utilisation. Of 74,722 admissions, 19.0% were insured by HICS. About 14.0% of the outpatient records were for HICS beneficiaries. Overall, the HICS utilisation rate in migrants was lower than in Thai patients. Being insured with the HICS significantly increased OP utilisation by 1.7%, and IP utilisation by 11.1% (relative to uninsured). Disease status was the most important factor that positively influenced the utilisation rate. Further studies that explore the differences in health service utilisation among HICS beneficiaries with diverse economic backgrounds are recommended

## 1. Introduction

Of the world’s 7.7 billion population, more than 258 million people (3.4%) live outside their country of origin [1]. This number is predicted to reach 405 million by 2050 [2]. Therefore, migrant health is a major part of global health policy discourse. 

Many national and international high-level meetings have focused on health protection for migrants including the United Nations General Assembly meeting in 2006, the World Health Assembly (WHA) Resolutions (WHA 60.26, WHA 61.17, and WHA 70.15), and the Global Compact for Safe, Orderly and Regular Migration adopted by the member states of the United Nations in 2018 [3,4,5,6,7]. The Sustainable Development Goals (SDGs) include migrants’ health as a part of Universal Health Coverage (UHC) in order to ‘leave no one behind’. [8]. Despite existing theoretical progress toward improving migrants’ health and social protection, implementation challenges remain on how to effectively translate the concept into real action and how to strengthen multi-sectoral collaboration on migrant health issues.

Thailand is an important migration hub in Southeast Asia; it is a place of transition and destination for numerous neighbouring countries especially Cambodia, Lao PDR, and Myanmar (the CLM nations) [9]. The accumulated number of cross-border migrants in Thailand is approximately 3–4 million people and the majority of them are CLM nationals and dependents who crossed the borders without valid travel documents [10].

Although Thai immigration laws indicate that illegal or undocumented migrants must be deported, the Thai government exercised mostly lenient measures and introduced a nationality verification (NV) policy for undocumented migrants [11]. This is due to the fact that these migrants are a key contributor to the Thai economy. The NV measure, in essence, tries to legalise undocumented migrants, allowing them to live and work in the country lawfully within a certain period (normally about four years). The NV migrants are required to acquire a work permit and buy health insurance, namely the Health Insurance Card Scheme (HICS), which was introduced in 2004 and managed by the Ministry of Public Health (MOPH). It allows migrants to enjoy public health services with a wide range of benefits, including ambulatory care, hospital admissions, and health promotion.

The HICS underwent a significant change in 2013 when the benefit expanded to cover antiretroviral treatment (ART) for HIV/AIDS. Additionally, the cost of the annual premium changed from 1300 Baht (US$ 39) to 2200 Baht (US$ 67) [12,13]. 

After the military coup in mid-2014, another significant change in the HICS occurred. The ‘One-Stop Service’ (OSS) policy was launched to expedite the registering process of undocumented migrants and facilitate the NV measure. The HICS premium was also reduced from 2200 Baht (US$ 67) to 1600 Baht (US$ 48) to attract more migrants to be enrolled in the insurance scheme [12].

Although there has been some research on migrants in Thailand [14,15,16], little is known about the utilisation of health services among HICS beneficiaries, especially undocumented migrants. Therefore, this study examined the outcomes of the HICS in relation to migrants’ use of health services at public health facilities. It is hoped that this study will help to extend the value and academic knowledge of public health research regarding migrant health in Thailand. 

## 2. Methods

### 2.1. Study Site 

Ranong province was purposively selected as the study site because it has the highest proportion of registered migrants to Thai citizens compared with other provinces in Thailand (Figure 1). It has a huge diversity of ethnic groups, including Thais, tourists, stateless people, and cross border migrants, mostly from Myanmar. The majority of these migrants work in the agricultural and fishery sectors [17].

### 2.2. Data Sources

Facility-based data, which recorded inpatient (IP) and outpatient (OP) attendance at public health facilities, were used. All visitor data from Ranong Provincial Hospital in Muang district, Kraburi District Hospital in Kraburi district, and two health centres (one in each district) between 2011 and 2015 were retrieved. These two districts were selected because they are the most densely migrant-populated areas in Ranong. The OP data were analysed from these four facilities whereas for IP utilisation, only Kraburi District Hospital’s dataset was used. This was because there were no admission beds in health centres and the IP data in Ranong Provincial Hospital were not complete due to problems with the electronic software of the recording system. The OP data in 2011 were discarded due to incompleteness and poor quality. Accordingly, the analysis of OP data was conducted from 2012 to 2015. Including data from this period enabled us to capture the change of service utilisation before and after implementation of the OSS policy in 2014 and the change of the HICS benefit in 2013. It is noted that normal newborn delivery records were excluded to prevent duplication of mothers’ admission. 

### 2.3. Data Analysis and Variable Management

The main outcome variable was the volume of use or the count of visits per person per year. The dataset in the original format was arranged in a per-visit fashion. Thus, individuals with multiple visits in a fiscal year needed to be linked by their unique identifier, namely, the hospital number (HN), before undertaking further analysis. 

The key independent variable was the insurance variable, coded 0 for uninsured migrants, 1 for HICS, and 2 for Thai patients. Note that, according to the Thai National Security Act [18], all Thai citizens are covered by the Universal Coverage Scheme (UCS), the main public insurance arrangement for Thai populations. For convenience, the authors use the term UCS patients for Thai patients from this point onwards. In natural experiment language, the HICS patients can be considered as ‘treatment’ whereas the uninsured migrants and the Thai UCS patients were ‘control_1’ and ‘control_2’, respectively.

The overall utilisation rate of HICS and UCS beneficiaries were presented in terms of number of visits per beneficiaries per year. The utilisation rate of the uninsured could not be determined because the exact number of uninsured (unregistered) migrants is not officially recorded. 

Then univariate analysis was performed by comparing the characteristics and utilisation rate among patients (HICS patients, uninsured and UCS patients). Chi-square test and Analysis of Variance (ANOVA) were applied. We also performed multivariable analysis to assess the effect of the HICS on utilisation, by taking into account the influence of all covariates (which will be explained later). In IP care, the outcome variable (admissions per person per year) followed Poisson distribution. In OP care, the outcome variable (visits per person per year) followed Negative Binomial (NB) distribution. As a result, we opted to use Poisson regression and NB regression OP care. A *p*-value of 0.05 was used as the cut-off to determine the statistical significance. In terms of variable management, the analysis accounted for the effect of other confounders/covariates all at once; namely, sex (male/female), age (years), disease status (International Classification of Diseases-10 (ICD-10) and Diagnosis-Related Group (DRG)), hospital-level (provincial hospital/district hospital/health centre), and domicile (proximity of patient’s home to hospital). In addition, we tested the goodness of fit of our models (Poisson regression and NB regression) with the Ordinary Least Square (OLS), which is commonly used in conventional analysis. We found that the results from the OLS and our models were quite similar. However, the OLS violated the homoscedasticity assumption as tested by Breusch–Pagan test. We then decided to present only the results from Poisson regression and NB regression.

To create more explanatory power, some variables were slightly modified. For instance, age was converted to age group (0–7, 8–15, 16–30, 31–60, and over 60 years) and domicile was converted to hospital proximity. (Any individual with an address located in the same district as the facility was coded 1 (proximity) and coded 0 (non-proximity) if otherwise). 

There was special treatment before analysis. For disease status, the DRG is the main indicator of disease severity measurement in IP care (catastrophic/non-catastrophic) [19]. We treated DRG as categorial variable. In routine IP care of the Thai health facilities, the hospital staff create a five-digit code where the fifth code shows the severity of ill-health (ranging from 1 to 5). In this case, the fifth code of DRG that equates two or above was identified as ‘catastrophic’ illness. For example, DRG 06100 means an admission for herniorrhaphy without complication in unilateral hernia while DRG 06103 refers to an admission for herniorrhaphy with complication in strangulated hernia. 

However, the DRG was not used in OP care. To remedy this limitation, ICD-10 on principal diagnosis was used instead. The ICD-10 was a unique identifier for each diagnosis. For example, A09 refers to diarrhoea of presumed infectious origin and B24 denotes HIV/AIDS. We grouped ICD-10 into three categories: 1 for non-specific diagnosis, 2 for Z-group diagnosis, and 3 for Ambulatory Care Sensitive Condition (ACSC) diagnosis. The Z-group diagnosis meant any admission with an ICD-10 starting with Z. This group of diagnoses covered a number of minor illnesses and disease prevention activities such as vaccination, medical counselling, and family planning. The ACSC was used as a proxy for severe diseases. Other diagnoses beyond the ACSC and Z-group categories were considered as non-specific OP diagnosis. The list of ICD-10 that were identified as ACSC is presented in Appendix A.

A time variable was also included in the analysis. It was coded as pre-OSS (2011-2014) and post-OSS (2015 and onwards). The year 2015 was used as a cut-off point because the year 2015 was the time when One-Stop Service (OSS) policy was fully launched, which imposed the registration of all undocumented migrants. Besides, an interaction term between disease status and insurance status was included in the model as well as an interaction term between time and insurance status. 

All calculations were performed by STATA software version 14 (serial number = 401406358220). Robust standard error was applied when assessing statistical significance.

### 2.4. Ethical Consideration

Ethical approval was obtained from the Institute for the Development of Human Research Protections in Thailand (IHRP 1778/2557). The presentation is done only for academic purposes without disclosing any individual information.

## 3. Results 

### 3.1. Inpatient Care

#### 3.1.1. Overview of the Data

In total, we acquired 111,725 records between fiscal years (FY) 2011 and 2015. The majority of the dataset was drawn from Ranong Hospital (92,925 records). There was a large volume of newborn admissions (about 20.0% of the entire data)—as shown in Figure 2. When excluding newborns, the mean age of the patients was 33.7 years (standard deviation (SD) = 23.1 years). The median age was 33.0 years (Interquartile range (IQR) =33.0 years). Females accounted for 57.8% of all patients. Details of descriptive statistics sorted by beneficiaries are presented later. 

The UCS was the most common schemes in all years among these three entitlements. More than half of overall admissions were UCS patients (≈67.0%), followed by the HICS (≈19.0%), and the uninsured (≈14.0%). Overall utilisation volume in Ranong Hospital was three times higher than in Kraburi Hospital, where the highest admissions were found in FY 2013 (see Figure 3).

#### 3.1.2. Utilisation Rate

The utilisation rate of Ranong Hospital was two times that of Kraburi Hospital in both beneficiary types. The utilisation rate of the HICs patients was lower than that of UCS beneficiaries, particularly in Ranong Hospital, whereas the IP utilisation rate of both insurance schemes almost equalled, except in 2013 and 2014, when the utilisation rate of insured migrants was marginally higher than the UCS utilisation rates.

There was a small increase in the utilisation rate of the HICS holders in 2013 and 2014. However, in 2015, the utilisation of the HICS at Ranong Hospital declined approximately 50.0% compared with the year before—as shown in Figure 4.

#### 3.1.3. Descriptive Statistics and Univariate Analysis

Overall, of 74,722 admissions (after excluding newborn admissions right after birth), the number of females was larger than males in all beneficiary types. The majority of HICS admissions were in the working-age groups with a mean age of approximately 31 years. Over 20.0% of uninsured admissions were from children under seven years old. The number of UCS admissions due to catastrophic illness was the highest (23.6%), followed by uninsured and HICS patients (15.4% and 13.7%, respectively). Most patients were admitted to the hospital located closest to their place of residence. About 20.0% of hospital beds in Ranong Hospital were occupied by HICS beneficiaries, while in Kraburi Hospital the figure was 15.0% (see Table 1)

#### 3.1.4. Multivariable Analysis

Based on the Poisson regression, findings revealed that the HICS had significantly increased overall visits per year by 1.7%, compared with uninsured patient visits; however, the HICS effect was still lower than the UCS on IP utilisation (+8.7%). The effect tended to be stronger when adding a history of severe illness (+19.3%), but it remained smaller than the UCS effect (+33.6%). Increasing age and residing near a facility significantly enhanced the number of admissions (see Table 2). 

### 3.2. Outpatient Care

#### 3.2.1. Overview of the Data

Of the total 1,251,797 records in the four-fiscal-year-period, the OP cases in all facilities were populated in working-age people, to a greater extent than IP cases. Age groups with a great volume of OP use were early childhood and late adults (Figure 5). After excluding newborns, the mean age was 34.2 years (SD = 22.2 years) and the median age was 32 years (IQR = 35 years). Females slightly outnumbered males (55.0% v 45.0%). More information about age and other demographic profiles are shown later in Table 3.

Most patients were insured with the UCS (78.0%), followed by HICS (14.0%), whereas 8.0% of migrant patients were uninsured. The OP utilisation volume of HICS and uninsured patients seemed to be stable over time in all facilities, whereas UCS patients’ OP usage increased constantly, especially at health centre level (see Figure 6). 

#### 3.2.2. Utilisation Rate

It was obvious that insured migrants’ OP utilisation rate was lower than that of UCS patients. The huge difference appeared in the latest year (2015) in a provincial hospital where the UCS utilisation rate was six times higher than that of HICS members.

In Kraburi Hospital, OP utilisation by UCS patients was twice higher than by insured migrants between FY 2012 and FY 2013. The gap was smaller in the following year, as service utilisation by insured migrants became slightly higher than UCS beneficiaries in FY 2014. 

The overall utilisation rate at the district hospital was higher than the provincial hospital in every study year (see Figure 7).

#### 3.2.3. Descriptive Statistics and Univariate Analysis

About 14.0% of total records (172,463 from 1,251,797 records) were OP visits by HICS patients. The average age of uninsured patients was the lowest compared with other beneficiary types. UCS patients with ACSC (26.0%) had the highest OP visits, compared with HICS patients (19.3%) and uninsured patients with ACSC (8.3%). Uninsured patients in the Z group had the largest utilisation volume followed by HICS and UCS patients (59.4%, 32.7%, and 27.3%, respectively). Approximately four-fifths of insured migrants had a place of residence close to the health facility. 

Overall, the utilisation volume of HICS patients was largest in Ranong Hospital (16.0%), followed by Kraburi Hospital (11.0%) and health centres (7.0%) (see Table 3).

#### 3.2.4. Multivariable Analysis

Regarding the NB regression, the effects of insurance in both the HICS and UCS per se led to increased frequency of OP visits by +9.9% and +33.6%, respectively, compared with uninsured migrants. The interaction between the HICS and having a history of Z group (severe diseases) diagnosis contributed to the increase in OP utilisation rate by about +90.8%. However, this interaction effect was smaller than the interaction between the UCS and ACSC (+11.8% v +28.5%). Increasing age led to the increase in service use. Similarly, residence-facility proximity led to a 20.0% increase in OP utilisation. The level of facilities also influenced utilisation volume; district hospital and provincial hospital yielded about 45.5%–52.8% increase in OP usage, relative to the health centres (see Table 4).

## 4. Discussion

### 4.1. Result Discussion

This study suggests that the volume of health service utilisation of both insured and uninsured migrants was lower than in the UCS. A small increase in the utilisation rate of the HICS patients was found in 2013 and 2014, and the utilisation rate at Ranong Hospital in 2015 declined about 50.0% compared with the year before due to an increase in the volume of registered migrants during the OSS era (increased denominator). The difference was more distinct for OP care and services in the provincial hospital. This phenomenon might be explained in certain ways. First, patients needing admission were more likely to have more severe diseases than those patients needing outpatient care, leading to the fact that migrant patients tended to show up at health facilities when they were seriously sick. This explanation was consistent with the Srithamrongsawat et al. study [20], revealing that OP visits of the UCS were about three times higher than those of the HICS, and 1.5 times larger than those of IP services. Some international literature also presents findings similar to this observation. For example, Shanmugasundaram et al. [21] suggested that most minority people were likely to be admitted to a hospital when they suffered from severe illness, compared to when they developed minor diseases. However, this finding must be interpreted cautiously. This is because univariate analysis shows that the proportion of UCS cases with severe diseases (catastrophic illness) was greater than the same proportion of cases in the HICS. Nevertheless, that does not mean that HICS patients visited a hospital with less severe conditions compared with UCS patients. To reach a conclusion on this point, it is necessary to conduct a household-based survey among all the beneficiary types (including the uninsured) to assess if there is a difference in the disease profiles among the ‘no-shows’. This is a type of ‘unmet need’ survey. A recent study by the International Health Policy Program (IHPP) [22] revealed that the unmet need for IP and OP services among urban refugees was higher than in the Thai population (28% vs. 0.4% and 54.1% vs. 1.4%, respectively) [23]. However, the population in the IHPP study differed from the population in Ranong, something which should be explored in a further study.

Secondly, the difference in utilisation rate between the UCS and the HICS was more obvious in Ranong Provincial Hospital, which is located in the headquarter district; whereas the difference was less pronounced in Kraburi district, which is a rural area. This might be explained by findings from a study by Suphanchaimat et al. [24] which revealed that better support from employers and peers in rural areas helped migrants get enhanced access to healthcare. Kraburi district is a special case, as migrants living there seemed to have a better economic status than those in the headquarter district where most migrants were living in poor urban settings [24]. Another explanation is the nature of work that might affect migrants’ access to health care. Most migrants in Kraburi district were agricultural workers, while those in the headquarter district were seamen who mainly spent time offshore, leading to difficulty in access to health services.

Thirdly, the greater utilisation rate of UCS patients relative to migrant patients could be ascribed to the ‘Healthy Migrant Effect’, which suggests that immigrants tend to have better health status than their native counterparts [25,26,27]. However, this might not be a strong explanation since, in multivariable analysis, we had already included key potential confounders, such as disease status, age, and sex in the model, and findings still showed that UCS patients used services more frequently than migrants did. 

Fourthly, despite not being the primary outcome, disease status contributed to the rise of OP visits, and its effect was more pronounced when interacting with the possession of health insurance. Both severe diseases (like ACSC diagnosis) and seemingly mild conditions (like Z-group diagnosis, which encompasses a vast range of health-promotion and disease-prevention activities) demonstrate positive effects on utilisation when interacting with both the UCS and the HICS (as presented in Table 4) and this effect was much larger than the effect of the insurance alone. A possible explanation for this is the HICS and the UCS equally provide comprehensive benefits, ranging from basic health promotion like vaccination to high-cost treatment, like caesarean section and HIV/AIDS treatment [28]. Such a phenomenon alludes to the fact that having public insurance leads to increase in utilisation only to a small degree if it is not co-driven by illnesses. This finding also contradicts some widespread social discourses, which claim that migrants are overusing public services and that insurance measures on migrants are somehow consuming public resources [29,30]. 

The positive effect of health insurance on service utilisation among migrants found in this study is supported by some international literature. Diaz-Perez et al. found that the community health service for Mexican immigrants in rural Colorado could help be effective in uncovering medical and mental illness and in directing patients to a health care home [12]. Another study in Germany pointed in the same direction. A cross-sectional survey of about 2000 migrants in Germany, conducted by Müllerschön et al., discovered that sub-Saharan African migrants in Germany who did not possess health insurance had lower odds of undertaking an HIV test than participants with health insurance [31]. However, we found that the effect of the size of the insurance variable on service use was small. This means that the presence of migrant health insurance alone does not guarantee increased utilisation of health services. If the policy aims to facilitate access to healthcare for migrants, other supporting mechanisms, such as migrant-friendly services, health education, and increasing trust between providers and migrant service users, must be in place [32,33]. 

### 4.2. Methodological Discussion

From a methodological point of view, this study has both strengths and limitations. In terms of strength, this study is one of the first in Thailand that explores individual records of those who present at the facilities. This approach, despite taking more time for data cleaning, allows us to access uninsured patients’ information, which is not normally found in the dataset of the central authorities. Additionally, using health facility data reaps benefits as this provides more numerous records of uninsured migrants compared with using data stored at the MOPH. This is because local facilities are not obliged (and it is not necessary) to submit the service records of uninsured migrants to the MOPH. Moreover, the data submitted to the MOPH are mainly for reimbursements, which are based on ‘insured’ records, in this case, the HICS. The way we analysed and utilised the data can serve as an example for other researchers to encourage the best use of existing real-world data to extend the value of public health research in relation to migrants.

Some limitations remain. First, we still face limited power to generalise as the analysis was confined to a single province only. Second, the data were limited to those showing up at the facilities. Patients’ data at the household level is lacking. This means data from patients who had not yet visited the facilities within the study period are discarded. The third concern is the lack of information for tracking the same individual across facilities. In this study, we used the hospital number as a unique identifier, but the hospital number in one hospital is not necessarily consistent with that found in another hospital. A more practical approach is using the national identification number (13-digit ID), but accessing the 13-digit ID is highly sensitive and always raises huge ethical concerns. Therefore, we could not explore whether the patients moved to other health facilities and if they did, where they went. However, it was estimated that the number of patients who changed facilities was low since HICS insurees must only visit the registered hospital. Patients who visit the facilities beyond the registration have to pay by themselves except for severe patients. According to Table 2, only 13.7% of HICS patients were patients with catastrophic conditions, and the possibility of referring those patients might be lower than that. Lastly, there are likely to be other factors that determine the use of services, but which are unobserved in this study. These include socio-economic variables, differences in culture and religious beliefs, length of stay in Thailand, perceived quality of care, fear of stigma and discrimination, and attitudes of the service providers. 

Future studies that delve deeper into these aspects are of great value. Additional studies on providers’ and migrants’ perceptions towards the HICS in the backdrop of the Thai healthcare system are likely to be helpful for further improvement of the policies to protect the health of migrants in Thailand. 

## 5. Conclusions

It is clear that the HICS helped increase service utilisation among migrants in both ambulatory care and inpatient treatment, relative to the services used by uninsured migrants. However, the HICS’ positive effect on service use was smaller than the effect of the UCS, the main insurance arrangement for Thai citizens, on UCS members. Disease severity was one of the main factors that contributed to the enhancement of service utilisation, and its effect was even larger than the insurance effect per se. Future studies that delve into the patients’ household level are recommended. Additional research that explores providers’ and migrants’ perceptions towards the HICS and the Thai healthcare system will be of great value to further improve policies to promote the health of migrants in Thailand.

## Figures and Tables

**Figure 1 ijerph-17-04431-f001:**
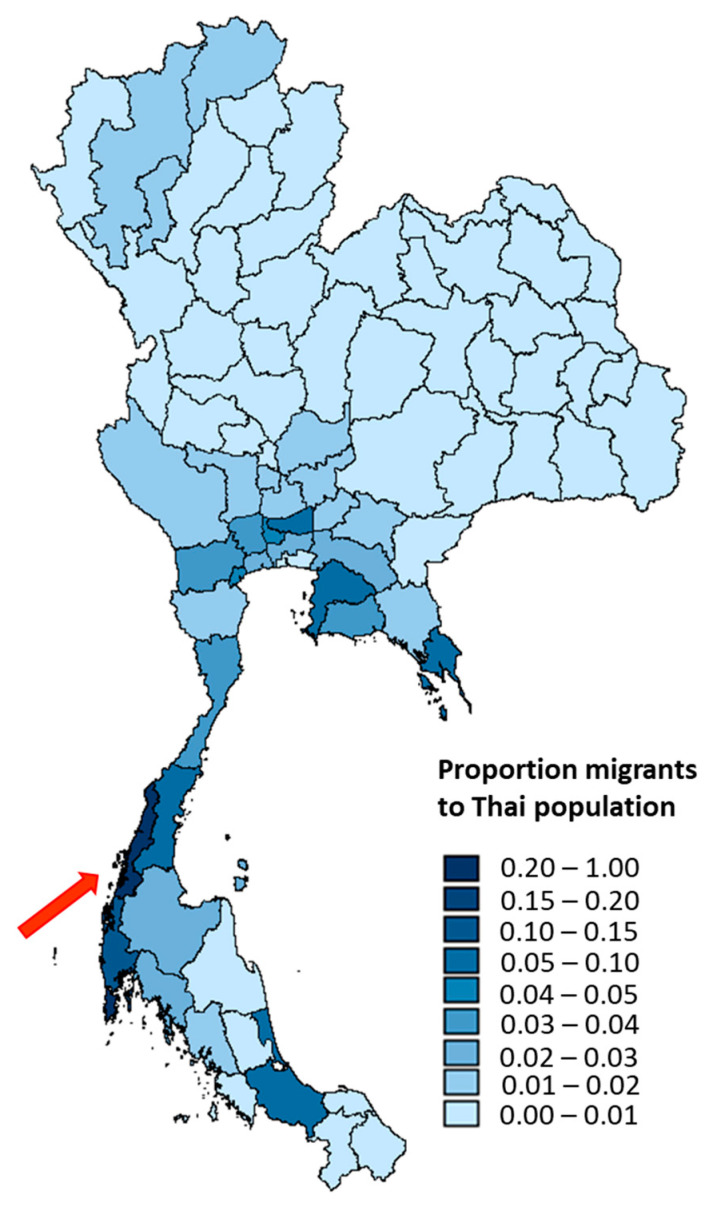
Ratio of registered migrants to Thai citizens in all provinces in Thailand.

**Figure 2 ijerph-17-04431-f002:**
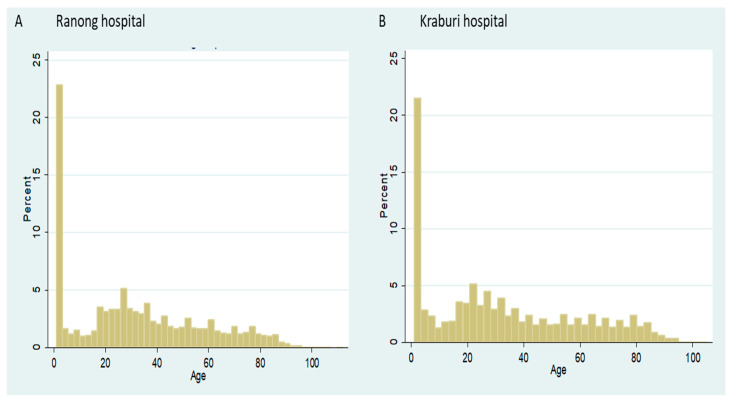
Percentage of inpatient (IP) utilisation volume in Ranong Hospital (**A**) and Kraburi Hospital (**B**) by age.

**Figure 3 ijerph-17-04431-f003:**
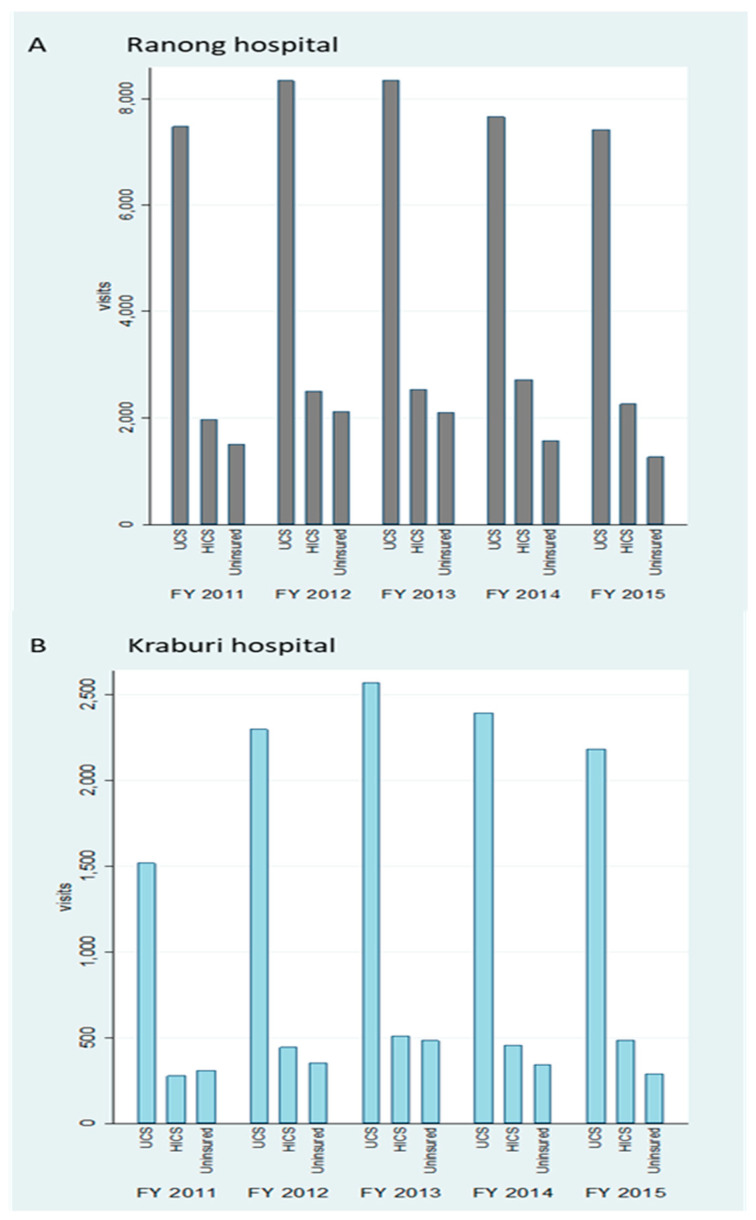
IP utilisation volume in Ranong Hospital (**A**) and Kraburi Hospital (**B**) by the top three most common insurance schemes by years.

**Figure 4 ijerph-17-04431-f004:**
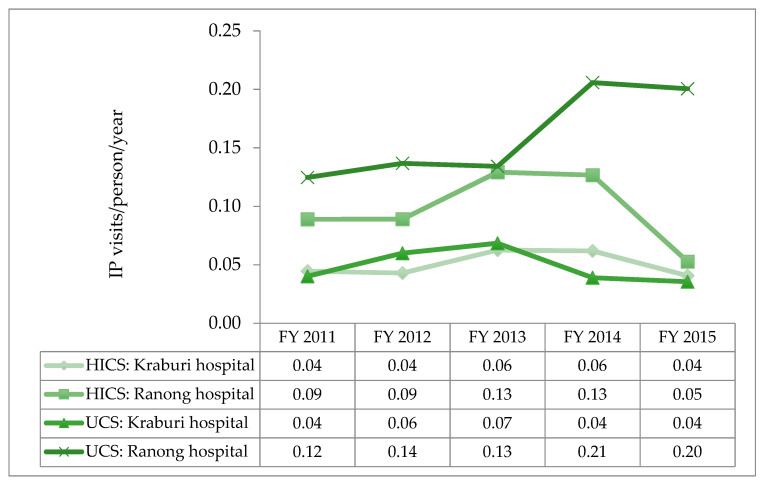
IP utilisation rates between the Health Insurance Card Scheme (HICS) and the Universal Coverage Scheme (UCS) beneficiaries by years.

**Figure 5 ijerph-17-04431-f005:**
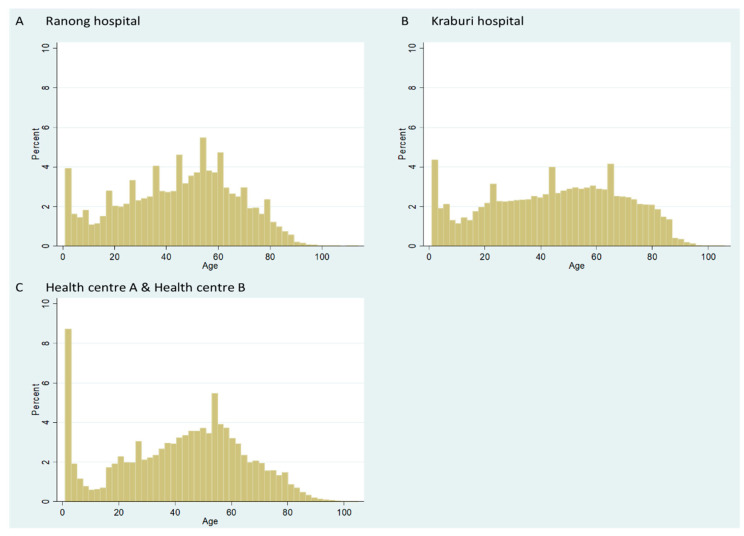
Percentage of outpatient (OP) utilisation in Ranong Hospital (**A**), Kraburi Hospital (**B**), and Health Centres A and B (**C**) by age.

**Figure 6 ijerph-17-04431-f006:**
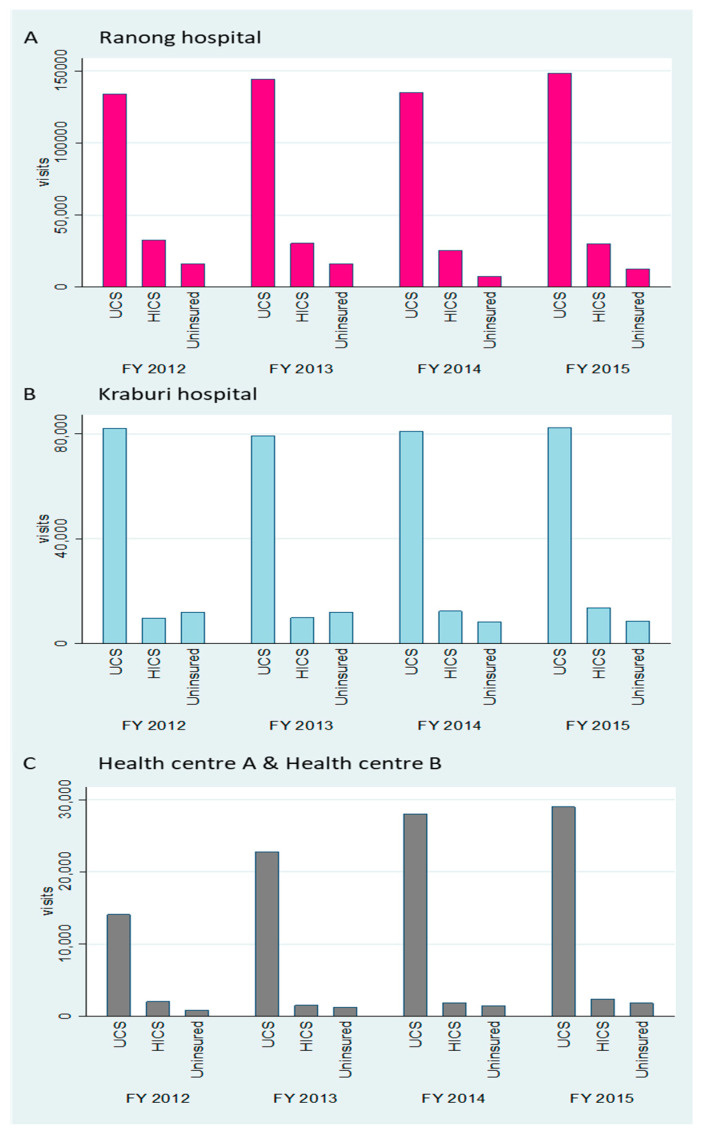
OP utilisation volume in Ranong Hospital (**A**), Kraburi Hospital (**B**), and Health Centres A and B (**C**) by the three most common insurance schemes across years.

**Figure 7 ijerph-17-04431-f007:**
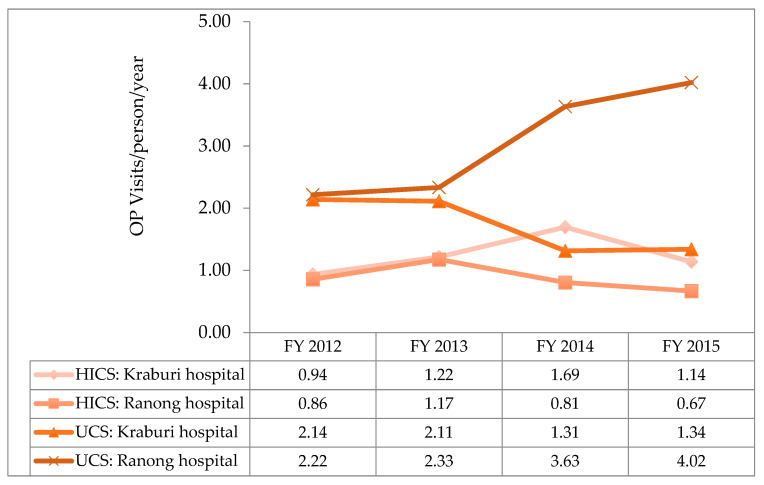
OP utilisation rate between the HICS and the UCS beneficiaries by years. Note: The utilisation rate at the health centre level cannot be analysed due to a lack of an accumulated number of registered migrants at subdistrict level.

**Table 1 ijerph-17-04431-t001:** Comparing total admissions by personal attributes and insurance schemes.

Variable	Uninsured N = 10,382	HICS N = 14,165	UCS N = 50,175	Test (*p*-Value)
Age—mean (SD)	26.2 (18.7)	30.9 (13.7)	39.2 (26.5)	ANOVA (<0.001)
Age group—n (%)				Chi-square (<0.001)
• =<7 y	2327 (22.4)	841 (5.9)	8302 (16.6)	
• 8–15 y	578 (5.6)	109 (0.8)	3468 (6.9)	
• 16–30 y	3700 (35.6)	6830 (48.2)	9651 (19.2)	
• 31–60 y	3215 (31.0)	5963 (42.1)	15,597 (31.1)	
• >60 y	560 (5.4)	422 (3.0)	13,156 (26.2)	
Female—n (%)	5935 (57.7)	9766 (68.9)	26,596 (53.0)	Chi-square (<0.001)
Female (non-delivery)—n (%)	4478 (51.7)	6612 (60.1)	24,466 (50.9)	Chi-square (<0.001)
Catastrophic illness—n (%)	1411 (15.4)	1722 (13.7)	10,392 (23.6)	Chi-square (<0.001)
Proximity to a facility—n (%)	6376 (63.4)	11,051 (83.6)	35,648 (71.1)	Chi-square (<0.001)
Facility level—n (% in a row)				Chi-square (<0.001)
• District hospital	1782 (11.9)	2172 (14.6)	10,974 (73.5)	
• Provincial hospital	8600 (14.4)	11,993 (20.1)	39,201 (65.5)	

Note: Missing data were negligibly small in number and were excluded from the analysis above. Post hoc analysis showed statistical significance with *p*-value of less than 0.001 for all matchings (UCS v HICS; HICS v uninsured; and uninsured v UCS) in all independent variables. For example, the mean age of HICS insurees was greater than uninsured migrants but less than UCS members. The proportion of patients with close facility proximity was largest in HICS insurees followed by UCS members and uninsured migrants.

**Table 2 ijerph-17-04431-t002:** Multivariable analysis of IP utilisation volume by the Poisson regression with interaction terms.

Variable	IRR	Std. Err.	*p*-Value	[95% Conf.	Interval]
Insurance (v uninsured)					
• HICS	1.017	0.005	0.001	1.007	1.026
• UCS	1.087	0.005	<0.001	1.077	1.096
Ever had catastrophic illness (v never)	1.057	0.012	<0.001	1.034	1.080
Insurance##Catastrophic illness					
• HICS##Ever had catastrophic illness	1.193	0.028	<0.001	1.140	1.249
• UCS##Ever had catastrophic illness	1.336	0.021	<0.001	1.295	1.379
Age group (v ≤ 7 yr)					
• 8–15	0.930	0.008	<0.001	0.914	0.945
• 16–30	0.965	0.006	<0.001	0.952	0.977
• 31–60	1.026	0.008	0.002	1.009	1.042
• >60	1.118	0.014	<0.001	1.092	1.145
Female (v male)	0.993	0.007	0.327	0.981	1.007
Proximity (v non-proximity)	1.106	0.007	<0.001	1.092	1.119
Post-OSS (v pre-OSS)	0.982	0.007	0.006	0.969	0.995
Insurance##OSS					
• HICS##Post-OSS	1.001	0.011	0.961	0.980	1.022
• UCS##Post-OSS	0.988	0.011	0.268	0.968	1.009
Provincial hospital (v district hospital)	0.997	0.009	0.711	0.980	1.014

Note: ## refers to interaction term.

**Table 3 ijerph-17-04431-t003:** Comparison of the sum of OP visits by personal attributes and insurance schemes.

Variable	Uninsured N = 99,119	HICS N = 172,463	UCS N = 908,215	Test (*p*-Value)
Age—mean (SD)	28.1 (17.9)	37.1 (17.4)	45.1 (23.9)	ANOVA (<0.001)
Age group—n (%)				Chi-square (<0.001)
• ≤7 y	17,560 (17.7)	8640 (5.0)	99,480 (10.1)	
• 8–15 y	4867 (4.9)	2967 (1.7)	62,978 (6.4)	
• 16–30 y	35,155 (35.5)	54,459 (31.6)	116,252 (11.9)	
• 31–60 y	36,865 (37.2)	89,975 (52.2)	405,526 (41.4)	
• >60 y	4668 (4.7)	16,421 (9.5)	295,974 (30.2)	
Female—n (%)	51,882 (52.4)	104,343 (60.5)	561,184 (57.6)	Chi-square (<0.001)
Disease status —n (%)				Chi-square (<0.001)
• ACSC	8266 (8.3)	33,313 (19.3)	254,508 (26.0)	
• Z group	58,901 (59.4)	56,412 (32.7)	267,653 (27.3)	
Proximity to a facility—n (%)	57,020 (61.5)	132,567 (80.6)	791,155 (82.8)	Chi-square (<0.001)
Facility level—n (row %)				Chi-square (<0.001)
• Health centres	5465 (5.1)	7913 (7.4)	93,973 (87.5)	
• District hospital	40,856 (9.9)	45,693 (11.1)	324,362 (78.9)	
• Provincial hospital	52,798 (7.2)	118,857 (16.2)	561,880 (76.6)	

Note: Missing data were excluded from the analysis. Post hoc analysis demonstrated statistical significance with *p*-value of less than 0.001 for all matchings (UCS v HICS; HICS v uninsured; and uninsured v UCS) in all independent variables.

**Table 4 ijerph-17-04431-t004:** Multivariable analysis of OP utilisation volume by the negative binomial regression with interaction terms.

Variable	IRR	Std. Err.	*p*-Value	[95% Conf.	Interval]
Insurance (v uninsured)					
• HICS	1.099	0.016	<0.001	1.068	1.130
• UCS	1.336	0.014	<0.001	1.309	1.364
Ever had ACSC (v never)	1.569	0.031	<0.001	1.510	1.630
Insurance##ACSC					
• HICS##Ever had ACSC	1.118	0.031	<0.001	1.060	1.179
• UCS##Ever had ACSC	1.285	0.027	<0.001	1.234	1.339
Ever had Z group (v never)	1.389	0.018	<0.001	1.355	1.423
Insurance##Z group					
• HICS##Ever had Z group	1.908	0.039	<0.001	1.832	1.987
• UCS##Ever had Z group	1.606	0.024	<0.001	1.560	1.653
Age group (v ≤ 7 yr)					
• 8–15	0.953	0.008	<0.001	0.936	0.969
• 16–30	1.017	0.009	0.043	1.001	1.034
• 31–60	1.471	0.013	<0.001	1.445	1.497
• >60	2.164	0.026	<0.001	2.114	2.216
Female (v male)	1.011	0.007	0.140	0.996	1.026
Proximity (v non-proximity)	1.200	0.012	<0.001	1.177	1.223
Post-OSS (v pre-OSS)	1.165	0.021	<0.001	1.126	1.206
Insurance##OSS					
• HICS##Post-OSS	0.864	0.019	<0.001	0.828	0.902
• UCS##Post-OSS	0.841	0.016	<0.001	0.811	0.872
Facility level (v health centre)					
• District hospital	1.528	0.015	<0.001	1.498	1.558
• Provincial hospital	1.455	0.014	<0.001	1.428	1.483
Alpha	0.438	0.005	-	0.428	0.448

Note: IRR = incidence rate ratio. ## refers to interaction term.

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
