# Peer review of "Outcomes of the Health Insurance Card Scheme on Migrants’ Use of Health Services in Ranong Province, Thailand"

_ijerph, 2020, doi:10.3390/ijerph17124431_

Round 1

Reviewer 1 Report

The manuscript is improved, but main issues like the statistical approach are not edited nor are the comments answered. My initially comments are attached as file and the unprocessed and also not answered comments are highlighted. Especially without working in the methodological statistic issues, I sadly cannot recommend this manuscript for publication.  

Reviewer 2 Report

The authors state that the purpose of their manuscript was "to examine the impact of the policy on the utilization rate of public health facilities among 14 HICS beneficiaries including undocumented migrants." To achieve this objective facility-based individual records between 2011 and 2015 were purposefully retrieved from one provincial hospital, one district hospital, and two health centers in one of the most densely migrant-populated provinces in Thailand. The results found by the authors suggest that: "Overall, the HICS utilization rate in migrants was lower than in Thai patients. Being insured with the HICS significantly increased OP utilization by 1.7%, and IP utilization by 11.1%, (relative to uninsured) Disease status was the most important factor that positively influenced the utilization rate. However, given the absence of a theoretical argument that assumes that the authors are making a relevant contribution to the literature, few adjustments are necessary for the article to be published, essentially for the following reasons:

  1. What is the contribution to the literature?
  2. What is the reason for studying this particular period (2011 to 2015)? And not another period?
  3. Highlight the managerial, practical, and political implications of the study.
  4. Table 3 and Figures 1, 3, and 4 are not cited in the text. It is important to write about them in the text for better understanding.
  5. The authors responded point by point to what was suggested by reviewers 1 and 2. In particular, I agree with what was suggested by the reviewers. The article has had significant improvements and may be considered for publication.

Round 2

Reviewer 1 Report

Now the manuscript is significantly improved. The applied methods are now comprehensible and the presentation of the results is adequate. I therefore propose to accept the manuscript in its present form for publication.

This manuscript is a resubmission of an earlier submission. The following is a list of the peer review reports and author responses from that submission.

Round 1

Reviewer 1 Report

General comments:

The authors face an interesting, and in times of worldwide migration and refugee flows important issue. While reading the paper, the question arises whether the authors have tried to distinct between refugees and migrants.

The use of the English language is very variable during the manuscript. The discussion is well written, clear and understandable. In contrast, the methods section and the results are difficult to understand for the reader due to the wording and sometimes grammatical errors. I recommend a double check by a native speaker after rewriting the manuscript.

The chosen statistical methods are another main issue, which has to be addressed, before this manuscript could be considered for publication.

Detailed comments are following below:

 1. Introduction

Line 55 – 57: please provide the information whether the price is per year or per month or singular.

Line 66: “will help to extend”

Line 67: “academic richness” is a very unusual idiom – please rephrase

2. Methods

2.1. Study site

Maybe a map would help the international reader to get familiar with the different destinations.

Line 70-74: please provide data and references for your statement

2.2. Data sources

Line 79: “retrieved. collected. “

Table 1 provides the same information as 2.2. Data sources. Please choose one form of presentation.

2.3. Data analysis and variable management

The description of the static methods is very bloomy and prosaic. But for a scientific publication he used language should be precise and clear. Unfortunately, the authors failed. Some examples are given below. Additionally, the rational for the selected statistical procedures (negative binomial regression and Poisson regression) is incomprehensible for the reader.

Line 97-98: the sentence is unnecessary, the abbreviation in brackets following the written-out idiom is commonly enough.

101-102: every statistical analysis contains a descriptive part and an analytical respectively inferential part. Please delete this sentence.

103-104: This is a little bit too much explanation for a publication in a scientific magazine. Most readers will know, how rates are calculated

104-106: This sentence is hard to understand due to grammatical issues. Please rephrase and keep it simple like: The utilization rate of the uninsured could not be determined because the exact number of the uninsured migrants is not officially recorded.

109-114: What I can understand from this paragraph, the authors tried some different kinds of regressions (which require different distribution of the outcome variable) and selected the ones with the best OLS parameter, “…. which is commonly used in most conventional analysis.” This is unfortunately the opposite of how it should be done. In a first step one should determine the distribution of the outcome variable, select the appropriate statistical procedure and run a null model. In the second step the independent variable gets included, the analysis is repeated and then the Akaike information criterion, the Bayesian information criterion and the OLS should show better that the predictor improves the model. The same should be applied to the included covariates. But at the end of the day, always the distribution of the dependent variable determines the kind of procedure.   

123-129 The explanation of the DRG variable is quite confusing. Maybe you should give an example. Did the authors treat the DRG variable as a continuous numeric variable or as a categorial variable? This is a very important to understand the applied model.

129-137: The same applies to the IDC-10 variable.

3. Results

3.1. Inpatient care

3.1.1. Overview of the data

Figure 1: please use the same y-axis units in both parts.

Figure 2: please use the same y-axis units in both parts or provide single figures. 

3.1.2. Utilization rate

A result section usually starts with the results of the descriptive statistic of the cohort (e.g.: age distribution, sex, newborns, adults, here regional distribution….)

Line 166: what effect is mentioned?

Line 168-169: belongs to discussion section

3.1.3. Descriptive statistics and univariate analysis

176-177: please check grammar

Table 2: Chi² Test for groups requires a post hoc test to provide useful information.  The present test does not tell the reader, which groups are compared and which ones showed a significant difference (age and hospitals). Column Variable should be left aligned.

3.1.4. Multivariable analysis

See comments to the statistical procedure.  The results are questionable due to the maybe wrong selected statistical procedure.

Table 3: Column Variable should be left aligned.

3.2. Outpatient care

3.2.1. Overview of the data

A result section usually starts with the results of the descriptive statistic of the cohort (e.g.: age distribution, sex, newborns, adults, here regional distribution….)

Line 196:  UCS ?

Line 200-201: please check grammar, nearly not understandable.

Figure 5: the histogram plots just give a rough overview. It would be better to provide a table containing statistical measure (min, max, median SD, SE, IQR….)

3.2.2. Utilization rate

3.2.3. Descriptive statistics and univariate analysis

Please see comments above

Table 4: please see comments to table 3

3.2.4. Multivariable analysis

Line 229-230: “…having a history of Z group (severe diseases) had a positive 229 effect on OP utilization rate by about +90.8%.” This main finding is only mentioned in the text. Perhaps the authors could illustrate this main finding by a figure?

4. Discussion

4.1. Result discussion

This section is written very good, clear and precise language is used, pleasant to read. Findings are set in context to relevant literature. Perhaps the reasons for the gap between the utilization of Ranong hospital by USC and others especially in 2015 (OP and IP) could be highlighted a little more.

4.2. Methodological discussion

Line 300-301: “Patients’ data at the household level is lacking. This means patients, who had not yet visited the  facilities within the study period, would be discarded.” It is not clear, what the authors mean. Patients who have not been admitted to an outpatient clinic or hospital haven’t even participated in the study…...?

Line 301-307: This is, beside the statistical issues, one main limitation of the study and should be not only named but broadly discussed. Please try to estimate the number and the possible influence of patients admitted to more than one facility during the study period.

Reviewer 2 Report

The paper looks into differential utilization rates of in-patient (IP) and out-patient (OP) care between migrants and non-migrants in a migration hotspot in Thailand, based on their health insurance status. All non-migrants are insured (covered by UCS scheme), while labour migrants are either registered, which comes with compulsory insurance (HICS scheme), or unregistered and thereby uninsured. The data record frequency of visits from two local hospitals and two health centres.

The set-up of the analysis is confusing, however. Utilization rates per group are calculated as “the number of visits divided by the volume of beneficiaries” (p.3), where it is indicated that this obviously cannot be calculated for the group of uninsured migrants, as the size of this population is by definition unknown, as these are not registered. This makes perfect sense, but then it is unclear how the uninsured can act as “control 2” for the insured migrants. Utilization rates simply cannot be compared, which explains why the uninsured group does not feature in Figure 3. However, the univariate and multivariate analysis again take on board this uninsured group, but the logic of ding so escapes me. What do we learn from this exactly?

Towards the end of the paper, where limitations of the study are discussed, it is mentioned that “data were limited to those showing up at facilities”, which raises the suspicion that the utilization rates are actually calculated differently from what was stated on p.3.

Hence, the paper requires fundamental reworking. Why not focus from the start on whether insured migrants make heavier (or lighter) use of healthcare than insured non-migrants? In the discussion part it pops up that the general idea is that migrants overuse public facilities (p.12), which would make an interesting opening hypothesis. Then a comparative two-group analysis could follow (where non-insured migrants are taken out), which would more effectively show that migrants tend to show up at a health facility later, i.e., in a more severe condition, so their frequency of healthcare seeking is lower, but if they do, it is more likely to require hospital admission. This could be nicely linked to the variation observed between hospitals, which in turn is plausibly linked to the type of work that migrants do (seamen vs. agricultural workers).

In the current version, however, the analysis does not seem to be guided by a clear question, nor does it provide a solid basis for the messages that are brought forward in the discussion part.